# Application of Palm Oil Biodiesel Blends under Idle Operating Conditions in a Common-Rail Direct-Injection Diesel Engine

**Ho Young Kim, Jun Cong Ge and Nag Jung Choi \***

Division of Mechanical Design Engineering, Chonbuk National University, 567 Baekje-daero, Jeonju-si, Jeollabuk-do 54896, Korea; jerryme@naver.com (H.Y.K.); freedefeng@naver.com (J.C.G.)
**\*** Correspondence: njchoi@jbnu.ac.kr; Tel.: +82-63-270-4765

**Abstract:** This study describes the effects of palm oil biodiesel blended with diesel on the combustion performance, emission characteristics, and soot morphology in a 4-cylinder common-rail direct-injection (CRDI) diesel engine. The operational condition is idle speed, 750 rpm (the lowest speed of the test engine without any operation by driver), and the load conditions of the engine are 0 Nm and 40 Nm. Five kinds of biodiesel fuels are blended with diesel in 0%, 10%, 20%, 30%, and 100% proportions by volume. A pilot injection was applied at BTDC 15 °CA and 20 °CA. Part of the pilot injection affects the combustion of the main injection due to the deterioration of the spray because of the high viscosity of palm oil biodiesel. Palm oil biodiesel is sufficient to keep the engine stable in an idling state, but the fuel economy deteriorated. The deterioration of the spray due to the high viscosity of palm oil biodiesel is offset by the effect of oxygen content and high cetane number, resulting in a constant nitric oxide ($NO_x$) emission. However, particulate matter (PM) is reduced. When the engine load is increased, the carbon monoxide (CO) emission amount increased because of the insufficient intake air and oxygen content to reduce the fuel-rich areas. However, when the palm oil biodiesel blend ratio was above a certain level, the influence of oxygen content in the palm oil biodiesel increased, resulting in reduced CO emission levels. Hydrocarbon (HC) was reduced by oxygen atoms in palm oil biodiesel. The sizes of particulates emitted from diesel engine using palm oil biodiesel decreased with an increased blend ratio because of oxidization of hydrocarbons absorbed on PM.

**Keywords:** palm oil biodiesel; idle; combustion; emission characteristics; soot morphology

## 1. Introduction

Fossil fuels are used in all areas of humanity and are primarily used as energy sources for internal combustion engines installed in automobiles, construction machinery, and ships. The consumption of fossil fuels has increased tremendously since their initial use. According to the research of US Energy Information Administration, 29% of the total energy is for transport purposes, and private transportation such as cars and trucks is highest [1,2]. This increase in the use of internal combustion engines causes problems such as exhaustion of fossil fuels, air pollution, global warming, and rising oil prices. Oil production is now at the limit of production [3]. In addition, regulations on environmental pollutants emitted from internal combustion engines have increased [4–6]. Accordingly, developments of both mechanical engineering technology and new fuels for internal combustion engines have been conducted to address environmental pollution and the depletion of fossil fuels. Policies have been put forward that require renewable energy [7]. A representative example of renewable energy is biofuels.

There are various biofuels such as biodiesel, bioethanol, and GTL (gas-to-liquid) fuel, which can replace energy sources in all transportation sectors [8–14].

Biodiesel is produced by converting triglyceride, which is the main component of vegetable oil or animal oil, into an alkyl ester by reacting with alcohol [15]. Since vegetable oil has a high viscosity, it cannot be applied to the engine without pretreatment. When it is changed into the alkyl ester form, the physical properties become similar to petroleum diesel and the cetane number increases. Therefore, it can be used without mechanically modifying a diesel engine [9,15]. To date, the biodiesel used worldwide has different main ingredients depending on the region. Examples of vegetable biodiesel include palm oil, jatropha, rapeseed, soybean, sunflower and coconut. Animal biodiesel and waste cooking oil are also used [9,16,17].

Many studies [18–20] have applied biodiesel blended with petroleum diesel to an internal combustion engine to study the performance and emissions of environmental pollutants. These studies show that applying biodiesel reduces the engine brake power and brake thermal efficiency (BTE) but increases brake specific fuel consumption (BSFC). In addition, hydrocarbon (HC), carbon monoxide (CO), carbon dioxide ($CO_2$) and particulate matter (PM) decreased, but nitric oxide ($NO_x$) increases slightly due to the characteristics of biodiesel, which affects combustion pressure, heat release rate. Also, studies on volatile organic compounds (VOCs), which are unregulated environmental pollutants emitted from engines using biodiesel, are also underway [21,22]. There is also active research on the morphology of soot from biodiesel and increasing interest in the fine dust generated [23,24].

In this study, palm oil biodiesel was selected among many biodiesels. It is known that palm has the highest oil content among raw materials of biodiesel, so its production cost is low. In addition, palm has the highest production rate among raw materials, and the process of converting to biodiesel is the same as using other raw materials. It is known that the physical properties such as viscosity, cetane number, and heating value of palm oil biodiesel are better compared to other type of biodiesel. In addition, the results of test using palm oil biodiesel on the real vehicles recoded much cleaner exhaust emissions [25]. A study of the combustion characteristics and environmental pollutant emission characteristics using palm oil biodiesel was conducted by Yusop et al. [26]; the engine power decreased, and the amount of PM emission decreased as the blend ratio of palm oil biodiesel increased. However, the $NO_x$ increased. Palm oil biodiesel has drawbacks such as low temperature fluidity which crystallize at low temperature as with other biodiesel; however, these advantages and research results show that palm oil biodiesel is a practical substitute for petroleum fuels.

In addition, the engine speed was set to the idling speed of 750 rpm. In the internal combustion engine, "idle" means that the engine operates at a rotational speed at which the engine generates a force to smoothly operate only essential parts such as a water pump, an alternator, a power steering pump, etc. without operating the accelerator pedal by a driver and the idle speed for passenger car is between 600 rpm and 1000 rpm and medium and heavy truck is almost 600 rpm [27]. Energy consumed by idling during vehicle operation was 17% in city driving mode and 4% on the highway [28]. As such, idling is a condition that needs confirmation to apply biodiesel in a real application. Many studies on the application of biodiesel have been carried out at engine speeds from 1500 rpm to 3500 rpm and at high engine load, not under idling conditions [19–21]. Many studies about the combustion and emission under idling condition have been using heavy trucks applying petroleum [29]. As the applicability to biodiesel increases, studies using various biodiesel under idling conditions were carried out [30–32]. Ashrafur Rahman et al. [31] performed their research at a fairly high idle speed of 1000 rpm to 2000 rpm. Roy et al. [32] was studied under high idle conditions of 1200 rpm, 1500 rpm, and 1800 rpm. However, the actual idle speed of the passenger car is set to be lower than 1000rpm.

To be more specific, this study was performed to investigate combustion and exhaust characteristics of a common-rail direct-injection diesel engine using palm oil biodiesel blends with high viscosity and oxygen characteristics under idling condition which has low injection pressure and low air movement in the cylinder.

## 2. Methodology

### 2.1. Test Fuels

Here, the blended biodiesel for test is expressed as PD (abbreviation for palm oil biodiesel). PD0 is 100% diesel with no palm oil blending, and PD100 is 100% pure palm oil biodiesel with no diesel fuel. The properties of diesel and palm oil biodiesel are shown in Table 1. There is no significant difference from the specifications of palm oil biodiesel used in other studies [26,33–35]. The density and viscosity of palm oil biodiesel is higher than diesel. But the caloric value of palm oil biodiesel is lower. The others, PD10, 20 and 30 are blended in 10%, 20% and 30% proportions by volume. With the specifications in Table 1 and references, as the blend ratio increased, the density and viscosity increased, but the caloric value decreased. The cetane index and oxygen content, which are the important characteristics of combustion and emissions, are going higher with increasing biodiesel blend ratios.

**Table 1.** Properties of test fuels.

| Properties | Diesel | Palm Oil Biodiesel | Test Method |
|---|---|---|---|
| Density at 15 °C (kg/m$^3$) | 836.8 | 877 | ASTM D941 |
| Viscosity at 40 °C (mm$^2$/s) | 2.719 | 4.56 | ASTM D445 |
| Lower calorific value (MJ/kg) | 43.96 | 39.72 | ASTM D4809 |
| Calculated cetane index | 55.8 | 57.3 | ASTM D4737 |
| Flash point (°C) | 55 | 196.0 | ASTM D93 |
| Pour point (°C) | −21 | 12.0 | ASTM D97 |
| Oxidation stability (h/110 °C) | 25 | 9.24 | EN 14112 |
| Ester content (%) | - | 96.5 Min | EN 14103 |
| Oxygen content (wt.%) | 0 | 11.26 | - |
| Sulfur content (wt.%) | 0.11 | 0.004 | ASTM D5453 |
| Hydrogen content (wt.%) | 13.06 | 12.35 | ASTM D5453 |
| Carbon content (wt.%) | 85.73 | 79.03 | ASTM D5291 |

### 2.2. Experimental Setup and Measurements

#### 2.2.1. Engine Setup

The engine used in the experiment is a four-cylinder 2.0-liter turbocharged-intercooled common-rail direct-injection diesel engine applied to a commercial vehicle. Detailed specifications are shown in Table 2. The FIE (fuel injection equipment, i.e., injector, fuel pump, common rail) and ECU (engine control unit) are Bosch systems.

**Table 2.** Engine Specification.

| Engine Parameters | Unit | Specification |
|---|---|---|
| Engine Type | - | In-line 4 Cylinder, Turbocharged, EGR |
| Maximum Power | kW/rpm | 83.5/4000 |
| Maximum Torque | Nm/rpm | 255/2000 |
| Bore × Stroke | mm × mm | 83 × 92 |
| Displacement | cc | 1991 |
| Compression Ratio | - | 17.7:1 |
| Number of Injector nozzle holes | - | 5 |
| Injector type | - | Solenoid |
| Injector hole diameter | mm | 0.17 |

#### 2.2.2. Experimental Equipment

As shown in the experimental equipment in Figure 1, an eddy current type dynamometer (DY-230 kW, Hwanwoong Mechatronics, Gyeongsangnam-do, Korea) equipped with a fuel supply system and a fuel pump was used. A multi-gas analyzer (MK2, Euroton, Italy) measured the levels of CO,

$CO_2$, and $NO_x$. HC was measured using a portable exhaust analyzer (HPC501, Pantong Huapeng Electronics, Shaanxi, China) [19]. A partial flow collecting type soot analyzer (OPA-102, Qurotech, Bucheon-si, Korea) was used to measure the PM level. The combustion pressure was acquired using a piezo-electric type pressure sensor (Kistler, 6056a, Winterthur, Switzerland) at the position of the glow plug and was recorded by a data acquisition (DAQ) board (PCI 6040e, National Instrument, Austin, TX, USA). The discharged soot was collected by a copper grid (FCF400-CU, Electron Microscopy Sciences, PA, USA) and was used to analyze the shape of the soot particles by transmission electron microscopy (TEM, H-7650, HITACHI, Fukuoka, Japan) and scanning electron microscopy (SEM, SUPRA 40VP, Carl Zeiss, Germany) images.

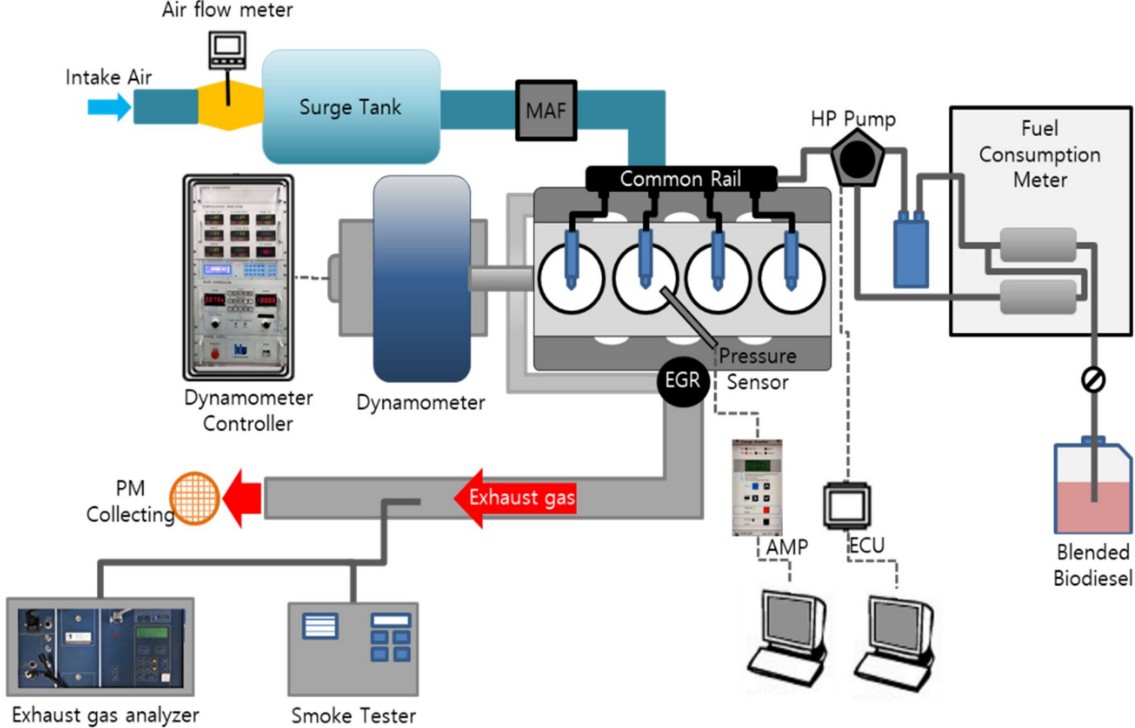

**Figure 1.** Schematic diagram of the experimental setups.

### 2.2.3. Test Procedure

In this experiment, the rotation speed of the engine was set to 750 rpm at idle speed as the ECU setting. The engine loads were 0 Nm, which is the no-load condition, and 40 Nm to simulate using an external load device on the actual vehicle. The auxiliary electrical losses, idle losses and parasitic losses on the vehicle were about 11% [36]. An uneven air-fuel ratio formed, PM formed in the rich region, and $NO_x$ is generated in the stoichiometric air-fuel ratio region because the diesel engine directly injects fuel into the combustion chamber at the end of the compression stroke. Many methods are used to reduce such PM and $NO_x$. Applying a pilot injection is a good way to reduce $NO_x$ without increasing PM [37]. Almost all the engines used in the current vehicle employ a pilot injection strategy. This test engine also employed pilot injection. The main injection timing was fixed at BTDC 2 °CA and pilot injection timings are BTDC 15 °CA and 20 °CA. The injection timings and durations by engine loads and pilot separations are shown in Figure 2. In addition, the exhaust gas recirculation (EGR) rate was set to zero to ensure stability of combustion in the idle state and to eliminate factors that could affect the combustion of palm oil biodiesel. The injection pressure was fixed at 280 bar. The experimental conditions are summarized in Table 3.

**Table 3.** Test Conditions.

| Test Parameters | Unit | Condition |
|---|---|---|
| Engine Speed | rpm | 750 ± 10 (Idle Speed) |
| Engine Load | Nm | 0 and 40 [1] |
| Cooling Water Temperature | °C | 85 ± 5 |
| Intake Air Temperature | °C | 25 ± 5 |
| Fuel Injection Pressure | bar | 280 |
| Injection Timing | °CA | Main BTDC 2/Pilot BTDC 15 and 20 |
| EGR rate | - | 0 |

[1] To simulate the use of external load devices of the actual vehicle such as air conditioners and electric devices.

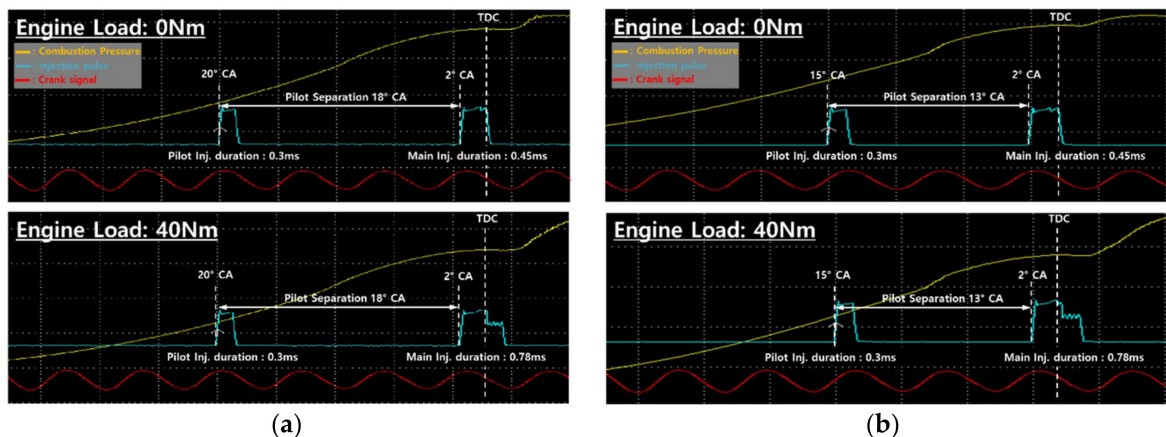

(**a**)                                            (**b**)

**Figure 2.** Injection timings and duration of pilot and main. (**a**) Main BTDC 2 °CA, pilot BTDC 20 °CA (separation 18 °CA), (**b**) main BTDC 2 °CA, pilot BTDC 15 °CA (Separation 13 °CA).

For each experimental fuel condition, combustion pressure and exhaust measurement were started when the coolant temperature was 85 ± 5 degrees and the engine speed was 750 ± 10 rpm with sufficient engine warm-up time and stabilization time. The combustion pressure is calculated as the average of 200 cycles. The sampling tube of the exhaust measuring instrument and the soot measuring instrument was directly inserted into the exhaust pipe. The analysis equipment for emission and PM which inhale the exhaust gas can lower the exhaust pressure from the engine can affect the engine power under low exhaust flow and pressure conditions, such as idling. Thus, combustion pressure, exhaust and soot measurements did not run at the same time. In addition, the emitted soot was directly collected from the exhaust pipe on the copper grid to measure the size and shape of the soot. The collected soot grids were evaporated in a 60-degree vacuum chamber for 12 h and images were taken at 100,000 times using TEM and at 250,000 times using SEM equipment.

### 2.2.4. Data Analysis

For each set of experimental conditions, the combustion pressure and heat release rate were averaged over 200 cycles of the engine running through the combustion analyzer. The heat release rate was calculated using the following formula [19–21,38]:

$$\frac{dQ}{d\theta} = \frac{k}{k-1}P\frac{dV}{d\theta} + \frac{1}{k-1}V\frac{dP}{d\theta} \tag{1}$$

Here, k is the specific heat ratio, P is the combustion pressure, and θ is the crank angle. The piston cylinder volume (V) can be calculated as a function of crank angle from the compression ratio, bore, stroke and length of connecting rod by sider-crank system. The volume of the cylinder (V) is given by Equation (2) as follows [38]:

$$V = \frac{V_d}{r-1} + \frac{V_d}{2}\left[1 + R - \cos\theta - \sqrt{R^2 - \sin^2\theta}\right] \tag{2}$$

where Vd is the displacement volume, r is the compression ratio and R is the stroke-to-bore ratio.

The specific fuel consumption (SFC) is calculated using the following formula [19,38]:

$$SFC = \frac{\dot{m}_f}{2\pi NT} \tag{3}$$

Here, $\dot{m}_f$ is the fuel flow rate, N is the engine speed, and T is the indicated torque calculated based on the indicated mean effective pressure (IMEP) for engine load 0 Nm or the brake torque from the engine dynamometer for an engine load of 40 Nm.

To check the combustion stability of the diesel engine operation at idle using each test fuel, the coefficient of variation (COV) for the indicated mean effective pressure was used, which is defined as [19]:

$$COV_{IMEP} = \frac{S}{X} \tag{4}$$

$$S = \sqrt{\frac{1}{N}\sum_{i=1}^{N}\{IMEP(i) - X\}^2} \tag{5}$$

$$X = \frac{1}{N}\sum_{i=1}^{N} IMEP(i) \tag{6}$$

Here, S is the standard deviation and X is the average of the IMEP over 200 cycles (here, N = 200), IMEP(i) is the indicated mean effective pressure (here, i = 1, 2, 3, . . . , 200).

Particle matter emitted from the diesel engine was collected on a copper grid, and the experiment for soot morphology and particulate analysis was carried out on a TEM and SEM. Particle agglomerates were measured using a SEM at 250 K magnification, and the acceleration voltage of the microscope was 10 kV. The TEM acceleration voltage was 100 kV, and the magnification was 100,000 times. The SEM images were modified using ImageJ software (http://imagej.net) to correct the boundary between soot particles to be more certain. The length of the soot particles was measured by randomly selecting a total of 200 particles. The average of the measured particles was calculated, and the distributions of particulate size were analyzed for each test.

## 3. Results

The engine performance, stability and fuel consumptions were evaluated for various test fuels, engine loads and pilot injection timings. At the same time, the levels of exhaust pollutants emitted in the gas state were evaluated. PM was collected on the grid and evaluated using TEM and SEM.

### 3.1. Engine Performance

#### 3.1.1. Combustion Characteristics

Typical properties of biodiesel (oxygen content in the fuel itself, high cetane number, a low calorific value, and the high viscosity) have a great influence on the injection and combustion in the cylinder. The mixture of air and fuel in the cylinder is the most unfavorable at low idle speeds, and the injection pressure is very low (fixed at 280 bar in this research). The development of fuel atomization in the cylinder deteriorated and the combustion efficiency decreased due to the high viscosity of the palm oil biodiesel. In addition, it is known that the surface tension is increased with increasing blend ratio. The high surface tension of the fuel as well as the influence of the high viscosity, also exacerbates the atomization [39,40]. In addition, the pilot injection time, 0.3 ms, is very short. Thus, the spread of the injection spray is not large, which affects the combustion.

Figure 3a shows the combustion pressure and heat release rate at an engine load of 0 Nm and a pilot injection timing of 20 °CA. The maximum combustion pressures are similar, but the heat release rate produced varied based on the combustion of pilot injections. The combustion starts at approximately the same point from the PD0 to the PD30 because of poor spray atomization by the high viscosity of palm oil biodiesel even though it has a higher cetane number. However, the PD100 starts the combustion slightly earlier, and the combustion becomes gentle. This suggests that the effects of oxygen content and high cetane number are more influential than the effect of high viscosity in PD100. Figure 3b reflects an engine load of 40 Nm under for a pilot injection timing of 20 °CA. The combustion pressure in the cylinder is higher, combustion starts at almost the same time, and the heat release rate rapidly increased. However, the peak heat release rate by the combustion of the pilot injection decreased as the blending ratio of palm oil biodiesel increased. The amount of intake air is the same, but the injected amount of the fuel increased, and the composition of the air-fuel ratio in the cylinder is not sufficient. The fuel injected at the pilot that has not been burned during the separation between the pilot and main injection is burned together with the main injection, and the heat release rate then becomes larger when the blend ratio of biodiesel is high. Figure 3c,d are the combustion pressures and the heat release rates for the engine loads of 0 Nm and 40 Nm under for a pilot injection timing of 15 °CA. Pilot injection is retarded 5 °CA. The temperature in the combustion chamber at the time of pilot injection, at 15 °CA, is higher than at 20 °CA, so the ignition delay period is shorter than for the previous pilot at 20 °CA. However, the mixing duration of the injected fuel and air becomes shorter. This poor mixture of fuel and air makes the heat release rate by the combustion of pilot injection lower. A higher blend ratio has a much lower heat release rate. At this duration of pilot combustion, the fuel of the pilot injection is not burned sufficiently and is burned with the main injection, resulting in a higher maximum heat release rate after the main injection.

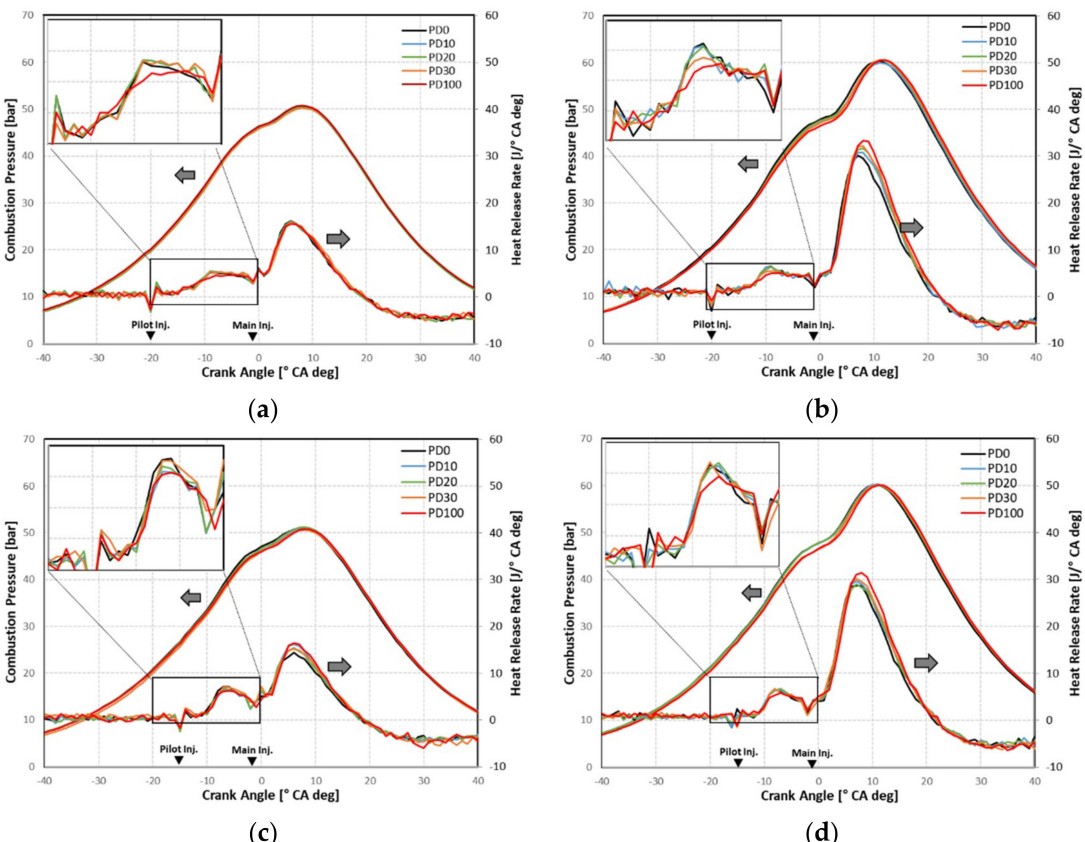

**Figure 3.** Combustion pressure and heat release rate. (**a**) Engine load 0 Nm and pilot BTDC 20 °CA, (**b**) engine load 40 Nm and pilot BTDC 20 °CA, (**c**) engine load 0 Nm and pilot BTDC 15 °CA, and (**d**) engine load 40 Nm and pilot BTDC 15 °CA.

As the blending ratio of palm oil biodiesel increases, the effect of maximum combustion pressure and maximum heat release rate due to pilot injection timing is reduced. Figure 4a,c,e show the plots of combustion pressure and heat release rate of PD0 (pure diesel), PD20, and PD100 (pure palm oil biodiesel) under an engine load of 0Nm at pilot injection timings 15 °CA and 20 °CA. In the PD0 and PD20, the maximum pressure due to the combustion of the pilot injection timing 15 °CA is higher than the result of 20 °CA, but the combustion pressure and maximum heat release rate in the PD100 at both pilot injection timings become similar. Figure 4b,d,f are the results of an engine load of 40 Nm. The maximum combustion pressure of 15 °CA is slower, and the heat release rate is higher but retarded in PD0. However, when PD100 is used, the maximum combustion pressure point and the heat release rate point and the level are similar. Deterioration of combustion due to the high viscosity of palm oil biodiesel reduced the effect of oxygen content.

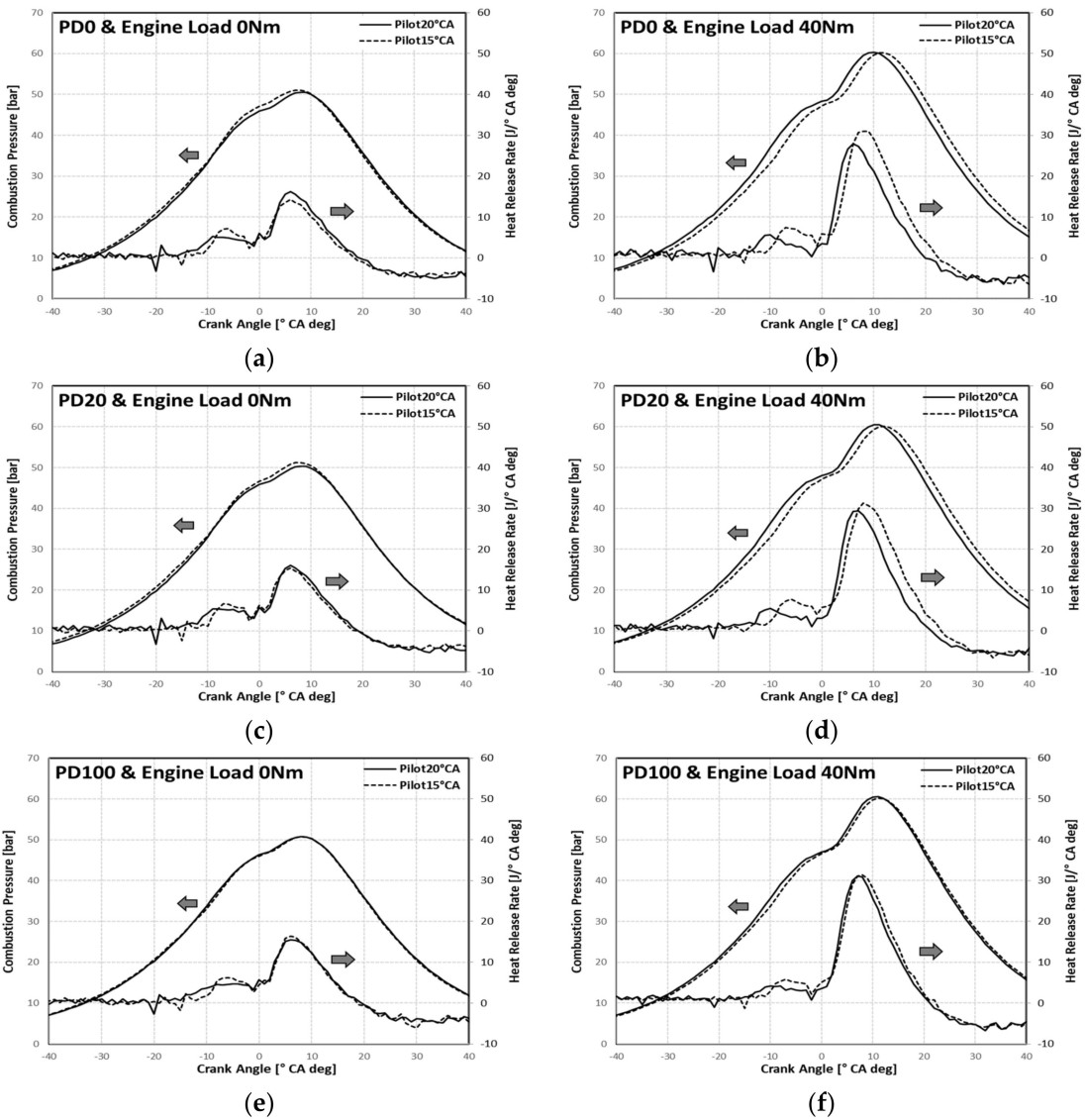

**Figure 4.** Combustion pressure and heat release rate by pilot injection timings. (**a**) PD0 and engine load 0 Nm, (**b**) PD0 and engine load 40 Nm, (**c**) PD20 and engine load 0 Nm, (**d**) PD20 and engine load 40 Nm, (**e**) PD100 and engine load 0 Nm and (**f**) PD100 and engine load 40 Nm.

### 3.1.2. Fuel Consumption

Since the brake torque is almost null at an engine load of 0 Nm, the BSFC cannot be calculated. Therefore, the indicated specific fuel consumption (ISFC) is calculated using the indicated torque calculated from the IMEP. At 40 Nm, the BSFC is calculated using the brake torque measured by the engine dynamometer. Figure 5 shows that the fuel consumption rate increases as the blend ratio of palm oil biodiesel increases. In general, the density of palm oil biodiesel is higher than that of diesel, which increases the mass per unit volume, and the lower heating value of palm oil biodiesel increases fuel consumption because it requires more fuel to produce the same output. The fuel consumption of pilot injection timing 15 °CA at an engine load of 0 Nm increased by 19% from 202 g/kW·h of PD0 to 248 g/kW·h of PD100 and increased from 214 g/kW·h of PD0 to 250 g/kW·h of PD100 at 20 °CA. At an engine load of 40 Nm, the pilot injection timing of 15 °CA increased by 12%, from 308 g/kW·h of PD10 to 350 g/kW·h of PD100. When the pilot injection timing is 20 °CA, the fuel consumption increased by 18% from 281 g/kW·h to 341 g/kW·h. In the case of an engine load of 0 Nm (a), the pilot injection timing of 15 °CA was burned with the main injection of a part of the pilot injection, the maximum combustion pressure increased, and the IMEP was higher. The position at which the maximum combustion pressure occurred was the same at both injection timings. Therefore, the ISFC at the pilot injection timing of 15 °CA was low. However, at an engine load of 40 Nm (b), the fuel consumption of 15 °CA at the pilot injection timing was as high as 8.8% for PD0, 8.3% for PD10, 6.2% for PD20 and 4.6% for PD30. As shown in Figure 4d,e, the maximum combustion pressure was generated at a position farther from the top dead center (TDC) during the expansion stroke period than the pilot injection timing 15 °CA, so that the efficiency for translation to power decreased. Furthermore, the faster injection timing indicates a longer the time to mix the fuel and the air, so the combustion efficiency increased. Therefore, as shown in Figure 4f, the small difference in combustion pressure and heat release rate in the PD100 can reduce the fuel consumption difference by 2.4%.

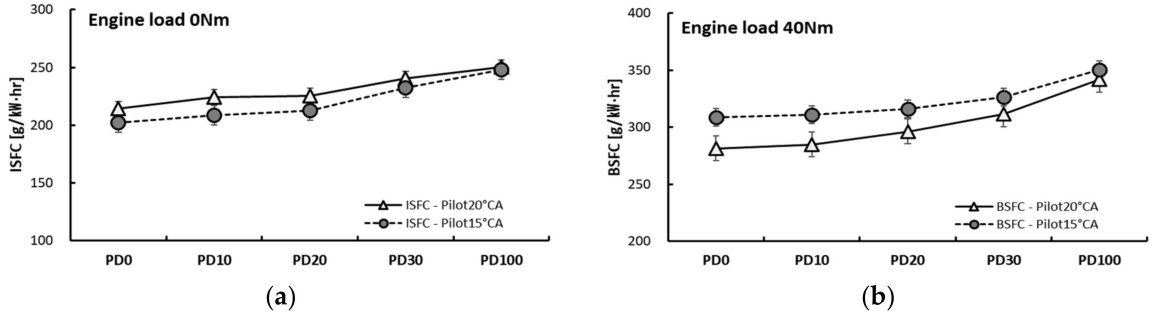

**Figure 5.** Specific fuel consumption: (**a**) ISFC of engine load 0 Nm (**b**) BSFC of engine load 40 Nm.

### 3.1.3. Idle Stability

Figure 6 shows the COV of IMEP and IMEP for various engine loads and blend ratios. When the engine load was 0 Nm using the PD100 fuel, the COV of IMEP was 3.3% in the case of a 20 °CA pilot injection timing. It was 3.5% in the case of 15 °CA. These results show that the engine operation is very stable. In PD10 and PD20, combustion is improved and stabilized for lower blend ratios. However, the increase of a part of the pilot injection that burned together with the main injection because of poor spray formation deteriorated the stability a bit when the blend ratio was over PD30. At an engine load of 0 Nm, the same tendency was observed in all the pilot injection conditions. However, a higher IMEP was obtained at 40 Nm when the pilot injection timing was 15 °CA. This may be due to the increase in the heat release rate and the retarded point of the peak combustion pressure by the retarded pilot injection timing.

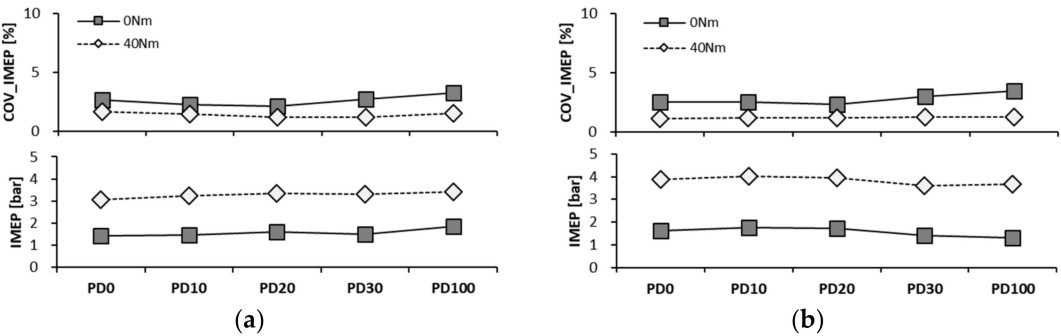

**Figure 6.** COV of IMEP and IMEP by pilot timings: (**a**) pilot BTDC 20 °CA, main BTDC 2 °CA, (**b**) pilot BTDC 15 °CA, main BTDC 2 °CA.

*3.2. Emission Characteristics*

3.2.1. Pollutant Emissions

As mentioned above, the diesel engine directly injects fuel into the combustion chamber at the end of the compression stroke, and an uneven air-fuel ratio is formed. Therefore, PM is formed in the rich region, and $NO_x$ is generated in the stoichiometric air-fuel ratio areas. It is generally known that the use of biodiesel improves combustion through the effects of oxygen, which increases $NO_x$ [18–20]. However, in the idle condition of this study, the deterioration of spray due to the high viscosity of biodiesel counteracts the reduced ignition delay due to the increase in cetane number and the combustion improvement effect due to oxygen content in the fuels itself. Figure 7 shows the result of $NO_x$ and PM. In the case of an engine load of 0 Nm and a pilot injection timing of 15 °CA, $NO_x$ was generated due to the rapid combustion of the pilot injection. In the case of an engine load of 40 Nm, the $NO_x$ was generated when the pilot injected fuel that did not burn during the separation between the pilot and main injection burned together with the main injection. PM decreased by reducing the local fuel excessive area and soot nucleus production due to the oxygen effect with an increase in the biodiesel blend ratio.

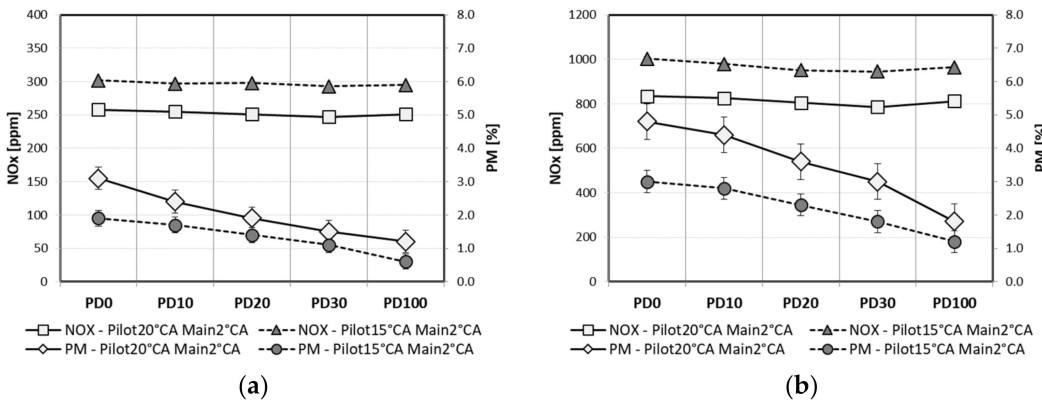

**Figure 7.** $NO_x$ and PM by engine loads: (**a**) engine load of 0 Nm and (**b**) engine load of 40 Nm.

The results of CO and PM are shown in Figure 8. The emission trend of CO differs depending on the engine load, while the emission pattern depends on the increase in the engine load. When the engine load is 0 Nm, the CO level decreased because the amount of intake air that can be mixed with the amount of fuel injected is sufficient, and there is suitable oxygen content in the fuel. However, as the blend ratio is increased (i.e., in the case of PD100) the deterioration of spray due to the higher viscosity of biodiesel makes more partially fuel-rich areas, so the level of CO increased. On the other hand, the CO level increased as the blend ratio increased for an engine load of 40 Nm. The injected fuel increased to an engine load of 40 Nm, but the amount of intake air is the same, so the fuel-rich

areas where CO is produced are not reduced by the effect of oxygen in biodiesel. However, when the blend ratio is much higher, the influence of oxygen increased, and the combustion pressure decreased. HC seems to be reduced due to the improved combustion caused by the presence of oxygen. Palm oil biodiesel contains oxygen in the fuel, which is more favorable than fuel and oxygen mixing conditions in the spray of pure diesel. At lower blend ratios, such as PD10 and PD20, HC levels of pilot injection at a timing of 15 °CA are lower than 20 °CA. However, at high blend ratios, such as PD100, the levels of 15 °CA are higher than at 20 °CA. The reasons for this are the pressure increase from combustion of the pilot injection and higher heat release rate of the main injection by poor spray development of the pilot injection at a pilot injection timing of 15 °CA. However, when the blend ratio increased, the combustion pressure and the heat release rate were similar to each other. The reduction of the HC is similar in these conditions, but HC is reduced because of the higher oxygen content in the fuel and the increased mixing duration for 5 °CA by the early pilot injection.

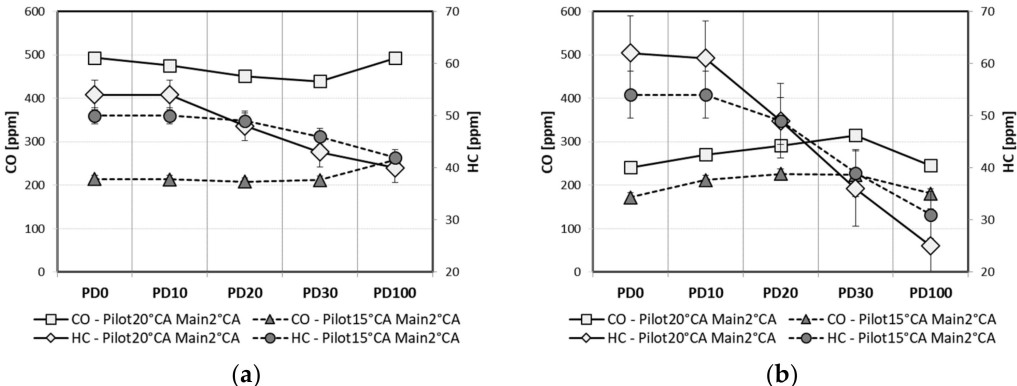

(**a**)                      (**b**)

**Figure 8.** CO and HC by engine load: (**a**) engine load 0 Nm, (**b**) engine load 40 Nm.

### 3.2.2. Particulate Matter Characteristics

The shapes and sizes of particulate matter discharged from the diesel engine are shown in Figures 9 and 10, which are TEM and SEM images. As shown in Figure 11b, 200 particles were randomly selected from SEM images, and their diameters were measured to analyze the average particle size. As shown in Figure 11a, the average particulate size decreased from PD0 33.9 nm to PD100 28.6 nm at an engine load of 0 Nm, and it decreased from PD0 34.3 nm to PD100 31.4 nm at an engine load of 40 Nm. The particulates are distributed from 15 nm to 60 nm, and most were between 25 nm and 45 nm. As the engine load increased, the amount of fuel injected increased, so the amount of PM and the sizes of particulates increased. As shown in Figure 12, the larger the engine load, the larger the distribution of PM particles. The soot emission pattern appeared to be partly clustered as the blend ratio increased, suggesting that the unburned hydrocarbons adsorbed on PM are oxidized by oxygen.

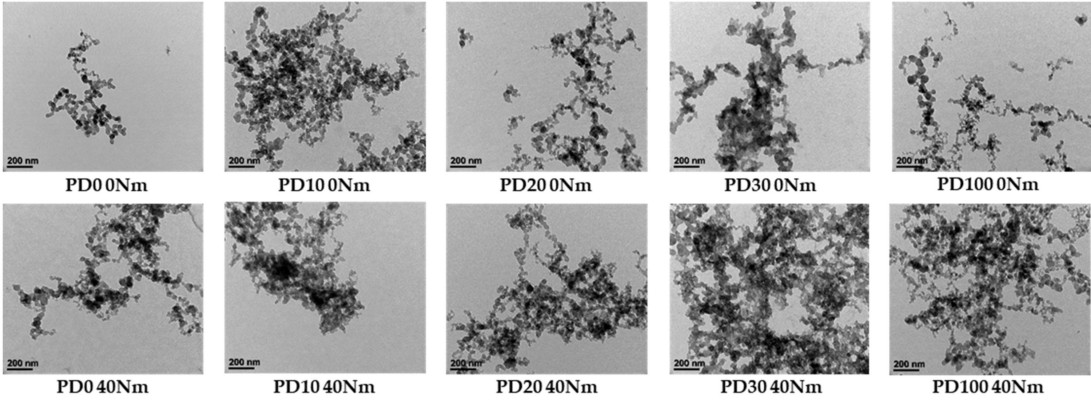

**Figure 9.** TEM images for various test fuels and engine loads (pilot BTDC 20 °CA).

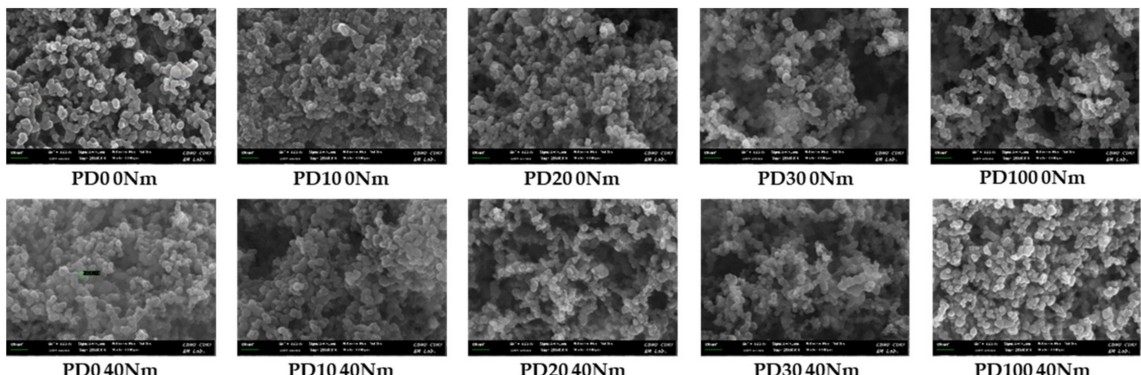

**Figure 10.** SEM images for various test fuels and engine loads (pilot BTDC 20 °CA).

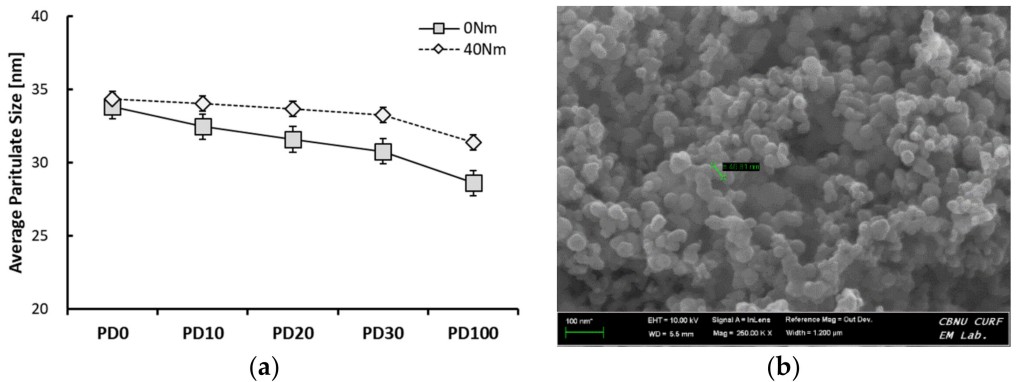

**Figure 11.** Average particulate size by engine loads: (**a**) average particulate size and (**b**) particulate size measurements.

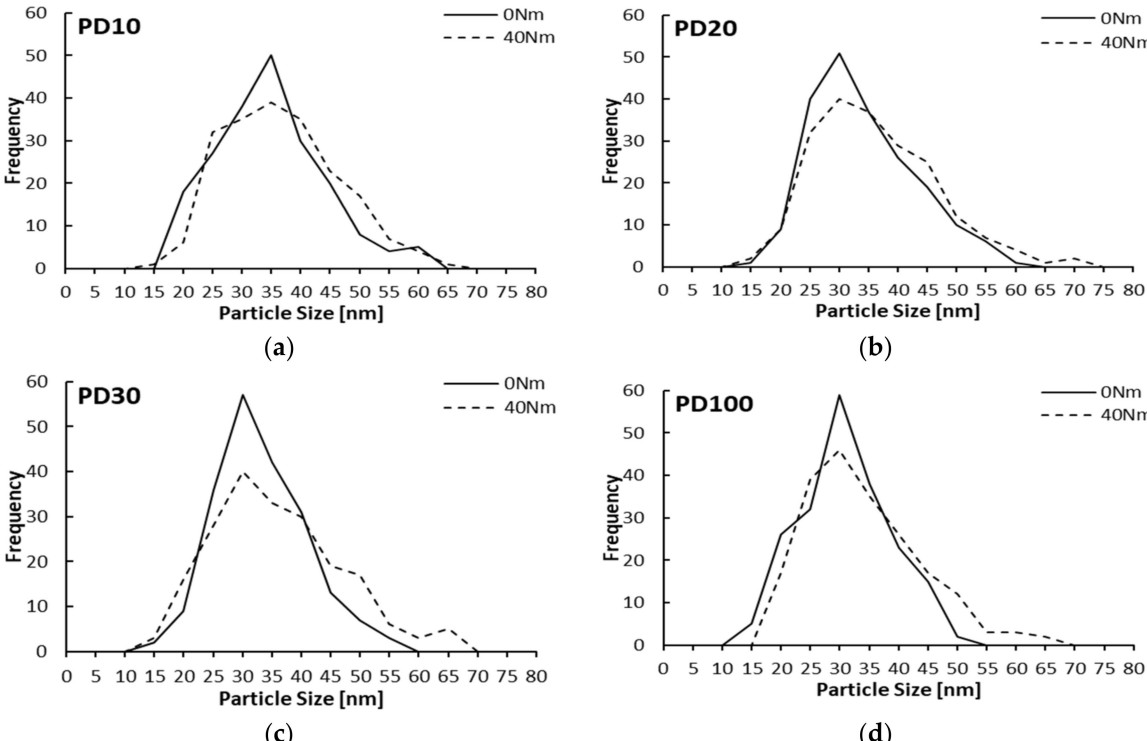

**Figure 12.** Particulate size distribution by test fuels and engine loads (Pilot BTDC 20 °CA): (**a**) PD10, (**b**) PD20, (**c**) PD30 and (**d**) PD100.

## 4. Conclusions

Under idle operational conditions with the lowest engine speed and lower injection pressure, the high viscosity of palm oil biodiesel reduces the atomization of the injected fuel, which affects the combustion more than the advantages of oxygen content and high cetane number. As the engine load increases, the combustion of the pilot injection deteriorates, and a part of the pilot fuel is burned together with the main injection, thereby increasing the maximum combustion pressure and the heat release rate. When the pilot injection timing is retarded, the maximum combustion pressure is farther away from the TDC, and the heat release rate is increased. However, as the blending ratio of palm oil biodiesel increases, the influence of the pilot injection timing is reduced. Fuel consumption increased with an increase in the blend rate. The stability of idling at all blend ratios is very good.

In the studies [30–32] where biodiesel was applied under idling conditions, $NO_x$ increased but PM, HC and CO decreased. However, these were evaluated above 1000 rpm, which is higher than the engine rotation speed applied in this study. It is considered that the idle speed lower than 1000 rpm and the low injection pressure give different tendency from the previous results. In result of this study, at low idling conditions, the deterioration of spray due to the high viscosity is offset by the effect of oxygen content and high cetane number, and the level remains constant. However, PM is reduced due to the reduction of the partial fuel surplus area by oxygenation. CO can be produced depending on the engine load. When the engine load is increased under idling conditions, the amount of fuel injected increases. However, under the conditions of low blend ratio, the fuel atomization of the injected fuel is worsened by the increased fuel, but the intake air amount becomes the same as that at the low load, which is not enough to reduce the rich region, and the CO emission increases. On the other hand, it is considered that the CO emission is reduced because the fuel injection amount is reduced at the lower engine load, and the influence of the oxygen is larger than that of the atomized fuel of the injected fuel is worsened. HC is reduced due to the influence of oxygen in palm oil biodiesel. The average sizes of the particulate matter decrease with increasing biodiesel mix. In addition, most of the particles are between 25 nm and 45 nm. As the fuel load increases, the distribution of larger particles increases. This result is similar to the study of Li et al. [23]. Furthermore, the emitted PM is partially clustered by the oxidation of unburned hydrocarbons.

Based on the results, palm oil biodiesel is sufficient as a fuel for diesel engines to provide power, stability, and reducing pollutant emissions when it is applied to diesel engines in idle condition. However, the fuel consumption is deteriorated. To overcome this disadvantage, it is necessary to study ways to the improve the fuel characteristics to lower the viscosity of the fuel while maintaining the advantages (i.e., oxygen content and high cetane number) while also strengthening the air flow in the cylinder. Systematic improvement is needed. An increase in $NO_x$ caused by the viscosity improvement of biodiesel under idling conditions can be sufficiently reduced by applying EGR.

**Author Contributions:** H.Y.K. suggested this research, performed the experiments, analyzed all experimental data, and wrote this paper. J.C.G. performed the experiments and contributed to the PM analysis. N.J.C. performed the data analysis and contributed to the discussion, and supervised the work and the manuscript. All authors participated in the evaluation of the data, and reading and approving the final manuscript.

**Funding:** This research was funded by [the Ministry of Education] grant number [2016R1D1A1B03931616].

**Acknowledgments:** This research was supported by the Basic Science Research Program through the National Research Foundation of Korea (NRF) funded by the Ministry of Education (Project No. 2016R1D1A1B03931616).

**Conflicts of Interest:** The authors declare no conflict of interest.

## Abbreviations

The following abbreviations are used in this manuscript.

| | |
|---|---|
| PD | Palm Oil Biodiesel |
| CRDI | Common-Rail Direct-Injection |
| BTDC | Before Top Dead Center |
| °CA | Crank Angle |
| $NO_x$ | Nitric Oxide |
| PM | Particulate Matter |
| CO | Carbon Monoxide |
| HC | Hydrocarbon |
| VOC | Volatile Organic Compound |
| GTL | Gas-To-Liquid |
| PD0 | 0% Palm Oil Biodiesel + 100% Diesel |
| PD10 | 10% Palm Oil Biodiesel + 90% Diesel |
| PD20 | 20% Palm Oil Biodiesel + 80% Diesel |
| PD30 | 30% Palm Oil Biodiesel + 70% Diesel |
| PD100 | 100% Palm Oil Biodiesel + 0% Diesel |
| COV | Coefficient of Variation |
| IMEP | Indicated Mean Effective Pressure |
| $dQ/d\theta$ | Heat Release Rate |
| k | Specific heat ratio |
| P | Combustion pressure |
| $\theta$ | Crank angle |
| V | Cylinder volume |
| $V_d$ | Displacement volume |
| r | Compression ratio |
| R | Stroke-to-Bore ratio |
| SFC | Specific Fuel Consumption |
| ISFC | Indicated Specific Fuel Consumption |
| BSFC | Brake Specific Fuel Consumption |
| $\dot{m}_f$ | Fuel flow rate |
| N | Engine Speed |
| T | Engine Torque |
| TDC | Top Dead Center |

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
