# Peer review of "Application of Palm Oil Biodiesel Blends under Idle Operating Conditions in a Common-Rail Direct-Injection Diesel Engine"

_applsci, doi:10.3390/app8122665_

Round 1
Reviewer 1 Report
Overall the manuscript is well written; the state of the art is adequately discussed even if the aim of the work should be better emphasized.
The description of the experimental apparatus is suffiecient; measurement procedures should be better detailed. The applied methodology for the evaluation of the particle size is not adequately explained and discussed.
The title of the paper is not fully representative of the work; “idle” is defined as lowest engine speed without load. The other investigated cases are useful for a deeper discussion of results but cannot be defined “idle conditions”.
Figures 7,9,10,13 _ error bars should be inserted; the accuracy in the measurement (and not instrumental sensitivity) should be discussed.
Most of the comments and conclusions are questionable. The effects induced in idle conditions by biodiesel at fixed fuel injection strategy are negligible and/or well known and expected. Starting from this point it is not clear the added value for literature of the proposed work.
Author Response
Thank you for your comments and suggestions concerning our manuscript. The comments and suggestions are all valuable and very helpful for revising and improving our paper, as well as the important guiding significance to researches. We have studied the comments carefully and have made the corrections. We hope it meets with your approval. The main corrections were marked in purple in the revised manuscript.
Please kindly find the attachment. Thank you.

Reviewer 2 Report
This paper presents a work that could be of interest to the reading audience of Applied Sciences. However, the introduction has not been made correctly. State of the art regarding palm biodiesel has not been appropriately revised. Some points should be revised before acceptance. My concerns are listed below.
The paper aims to study the combustion and emissions using palm oil biodiesel.
GENERAL
1. Title. The title of the article should be changed to: ‘Application of Palm Oil Biodiesel Blends under Idle Operating Conditions in a common-rail direct injection Diesel Engine’. Because ‘With Diesel’ is not necessary and because it is better to avoid acronyms in the title.
2. List of symbols and acronyms. Please add all symbols and acronyms that the authors use in the equations, tables, and text.
3. Introduction.
3.1. The introduction is short, and they have much well-known superfluous information that is not necessary for a journal of this level. It is necessary to profoundly change the first part (lines 31 to 78) of the introduction (please, see specific comments).
3.2 In the introduction, it is necessary to develop of the advantages and disadvantages in the palm oil biodiesel, the methods for the production of palm oil biodiesel and the emissions and benefits in the engines of palm oil biodiesel. There is much specific literature on the use of palm biodiesel. The authors should focus the discussion on the bibliographic review of the use of palm biodiesel to get more depth in the subject and be able to know more clearly the contribution that the document makes to the field of research.
4. Results. The authors must compare the results with those obtained by other authors.
SPECIFIC
5. Line 31. Please remove this sentence: ‘Civilizations developed rapidly as the use of fossil fuels began.’
6. Line 34. Please remove Figure 1 and this sentence: ‘Figure 1 shows the percentage of US energy use by sector’. This figure is not necessary to justify the distribution of the use of energy in the US, because it is well known. If the authors deem it convenient, modify the sentence 'As shown, 29% of the total energy is for transport purposes, and private transportation such as cars and trucks is highest [1,2].' (although it does not contribute anything to the theme of the article).
7. Line 39. Please remove Figure 2. This figure is not necessary; the information is well known in the field of study. Besides, it only corresponds to the European emission regulations (and previously they have developed the distribution of energy in the USA). This has no sense.
8. Line 44. It is recommended that the authors read the introduction of the following articles:
• D. Cárdenas, O. Armas, C. Mata, F. Soto, Performance and pollutant emissions from transient operation of a common rail diesel engine fueled with different biodiesel fuels, Fuel, Volume 185,2016, Pages 743-762)
• Lapuerta M, Armas O, Rodríguez-Fernández J. Effect of biodiesel fuels on diesel engine emissions. Prog Energy Combust Sci 2008;34:198–223
9. Line 45. Please remove this sentence: ‘Minnesota in the United States required use biodiesel in 2005 for the first time and announced a plan to mandate B20, which will increase biodiesel content from 10% to 20% in May 2018 [8]’. There are hundreds of initiatives of this kind all over the world. Besides, it was not the first US state to announce such plans. The authors should consult the plans of European cities if they want to give an example, although this type of information is known and unnecessary.
10. Line 55. Please remove these sentences: ‘Biodiesel was first studied in 1853 by E. Duffy and J. Patrick on the transesterification of vegetable oils. Then in 1893, Rudolf Diesel used peanut oil (vegetable oil) to operate a diesel engine. The initial diesel engine was made to operate using vegetable oil, but the use of petroleum diesel increased as the use of fossil fuels, and refining technology was developed [15].’. They are well-known.
11. Line 63. This sentence: ‘Examples of vegetable biodiesel include palm oil, jatropha, rapeseed, soybean, sunflower and coconut. Animal biodiesel and waste cooking oil are also used [9].’ Please add one or several references that serve to support that premise. For example:
• Niraj Kumar, Varun, Sant Ram Chauhan, Performance and emission characteristics of biodiesel from different origins: A review, Renewable and Sustainable Energy Reviews, Volume 21, 2013, Pages 633-658.
• Syed Ameer Basha, K. Raja Gopal, S. Jebaraj, A review on biodiesel production, combustion, emissions and performance, Renewable and Sustainable Energy Reviews, Volume 13, Issues 6–7, 2009, Pages 1628-1634,
12. Lines 66 to 70. Please modify the paragraph (lines 66 to 70) to include only results corresponding to palm oil biodiesel.
13. Line 67. This sentence: ‘These studies (what studies?) show that applying biodiesel reduces the engine brake power and brake thermal efficiency but increases brake specific fuel consumption. Hydrocarbon (HC), carbon monoxide (CO), carbon dioxide (CO2) and particulate matter (PM) decreased, but nitric oxide (NOx) increases slightly’. Please rewrite and add these studies.
14. Line 75. This sentence: ‘Studies on VOCs, which are unregulated environmental pollutants emitted from engines using biodiesel, are also underway [19].’ ‘Studies’, but there is only one reference, and it is not a revision. There are many articles on emissions from palm oil biodiesel. For example:
• George Karavalakis, Stamoulis Stournas, Evangelos Bakeas, Light vehicle regulated and unregulated emissions from different biodiesels, Science of The Total Environment, Volume 407, Issue 10, 2009, Pages 3338-3346.
15. Section 2.1. This sentence: ‘The specifications of fuels with the same blend ratio used in this experiment are shown in Table 1.’ Are these specifications from other studies? Alternatively, have they calculated from the compositions? Please do not mix the measured data with the data obtained from the bibliography. Indicate the data that comes from the bibliography.
16. Section 2.1. Please add the test method used to determine the properties of fuels.
17. Section 2.2.1. Does the used engine have EGR? What is the EGR rate used? Is the injector piezoelectric or solenoid?
18. Section 2.2.1. Please add the specifications of the exhaust gas analyser or add this reference:
• Jun Cong Ge, Min Soo Kim, Sam Ki Yoon, Nag Jung Choi, Effects of Pilot Injection Timing and EGR on Combustion, Performance and Exhaust Emissions in a Common Rail Diesel Engine Fueled with a Canola Oil Biodiesel-Diesel Blend. Energies 2015 8(7):7312-7325
19. Abstract, line 15: ‘A pilot injection was applied at BTDC 15°CA and 20°CA’. Section 2.2.1., Table 2... ‘BTDC7/ATDC43/BTDC52/ATDC6’. Why? To what operation condition corresponds?
20. Figure 4. Please, improve this figure. It looks blurry. The legend of the graph is impossible to read.
21. Line 156. This sentence: ‘The heat release rate was calculated using the following formula:’ Please add a reference.
22. Figure 5. Please, improve this figure. The symbols used to represent the series are tiny. Please change the colour code; it is impossible to distinguish the series represented.
Author Response

(The authors gave the same response as above.)

Reviewer 3 Report
This study investigates the effects of palm oil biodiesel (PD) blended with diesel on the combustion performance, emission characteristics and soot morphology in a diesel engine. Although the results seem to be reasonable, this manuscript cannot be accepted in its present form. Review comments are as follows.
1. Ultimate analysis and proximate analysis should be provided for diesel and biodiesel fuels.
2. Table 1 presents fuel properties. However, fuel properties used in this study should be different from those in the literature. Why should References [23, 27-29] be cited in this study? How do you obtain the fuel properties?
3. The authors stated that “The development of fuel atomization in the cylinder deteriorated and the combustion efficiency decreased due to the high viscosity and the high surface tension of the palm oil biodiesel.” However, the values of surface tension for different test fuels are not provided.
4. What are air-fuel ratios for the test conditions?
5. Fig. 10(a) shows a greatest CO emission for PD100, while Fig. 10(b) illustrates a lowest CO emission for PD100. Why? It is very unclear.
6. In the text, Figs. 9 and 10 are not mentioned.
7. Why is the particulate matter (PM) level in unit of %?
Author Response

(The authors gave the same response as above.)

Round 2
Reviewer 1 Report
Most of the remarks are adequately addressed. The paper can be approved for the publication.